# Predicting Macroinvertebrate Responses to Water Abstraction in Alpine Streams

Gabriele Consoli [1,2,3], Fabio Lepori [1], Christopher T. Robinson [2,3] and Andreas Bruder [1,4,*]

[1] Institute of Earth Sciences, University of Applied Sciences and Arts of Southern Switzerland (SUPSI), Campus Mendrisio, 6850 Mendrisio, Switzerland; gabriele.consoli@eawag.ch (G.C.); fabio.lepori@supsi.ch (F.L.)
[2] Department of Aquatic Ecology, Swiss Federal Institute of Aquatic Science and Technology (Eawag), 8600 Dübendorf, Switzerland; christopher.robinson@eawag.ch
[3] Institute of Integrative Biology, ETH-Zürich, 8092 Zürich, Switzerland
[4] Institute of Microbiology, University of Applied Sciences and Arts of Southern Switzerland (SUPSI), Campus Mendrisio, 6850 Mendrisio, Switzerland
[*] Correspondence: andreas.bruder@supsi.ch; Tel.: +41-586-666-222

**Abstract:** Exploitation of hydropower potential in alpine areas undermines the ecological integrity of rivers. Damming and water abstraction substantially alter the physical habitat template of rivers, with strong repercussions on aquatic communities and their resources. Tools are needed to predict and manage the consequences of these alterations on the structure and functioning of macroinvertebrate communities and resource availability in alpine streams. We developed habitat preference models for taxa, functional feeding guilds, and organic resources to quantify the effects of discharge alteration on macroinvertebrate communities in two alpine streams. Our physical habitat model related an indirect measure of bottom hydraulic forces (FST hemispheres) to the distribution of macroinvertebrate taxa and their resources. We observed that flow-dependent habitat availability for macroinvertebrate communities generally decreased with increasing water abstraction. We were able to relate these changes to near-bed hydraulic conditions. Our results suggest, however, the existence of upper discharge thresholds delimiting optimal habitat conditions for taxa. In contrast, we found weak effects of near-bed hydraulic conditions on resource distribution. Overall, our findings contribute towards predicting the impacts of water abstraction on macroinvertebrate communities in small alpine streams and the benefits of baseflow restoration.

**Keywords:** hydropower; physical habitat models; preference models; invertebrate traits; functional feeding guilds

## 1. Introduction

The hydropower potential of alpine streams is widely exploited for electricity generation. This involves the construction of dams for high-head storage and the development of a network of conduits for water abstraction, trans-catchment transfer and water return to downstream rivers after turbination. The resulting impacts range from baseflow modification to reductions in flow variability, often causing interruption of the river continuum [1,2] and alterations to natural sediment regimes (e.g., retention, flushing) [3,4]. The mechanisms and magnitude of downstream impacts depend on factors related to sediment and water availability, which can cause drastic changes in habitat conditions relevant for local biota [1]. The ongoing intensification of hydropower utilization as an alternative to fossil fuel use, in combination with the effects of climate change [5], will further increase pressures on stream ecosystems globally. In the coming years, the strategic role of hydropower will be consolidated, new hydropower infrastructure will be created, and concessions for hydropower operators will be re-negotiated [6]. These developments not only concern large and conspicuous hydropower schemes, but also, increasingly, those on small streams [7].

Managing the protection and multiple uses of alpine streams requires tools that relate hydrological alterations to ecological impacts. Such tools need to quantify relationships between water quantity, timing, and quality, with suitable indicators describing the aquatic biota [8]. They also need to predict the extent by which planned hydropower schemes will impact river ecosystems and biota, and how hydro-morphological restoration projects (e.g., environmental flows) can improve habitat conditions in regulated rivers [9]. To achieve the best compromise for water allocation, restoration projects need defined ecological goals based on mechanistic understanding of relationships among flow, sediment, and biota [10]. It is thus necessary to quantify the effects of changes in discharge on benthic habitat availability and quality, and how these affect river ecosystems, aquatic organisms, and in turn, ecosystem functions and services.

Physical habitat models are widely used to quantify relationships between descriptors of near-bed hydraulic conditions (e.g., shear stress, Froude number, or combinations of water velocity, depth, substrate composition) and aquatic organisms, mostly fish and macroinvertebrates. These models typically couple a hydraulic model, predicting the effect of discharge on instream hydraulic characteristics, and a preference (biological) model, predicting the effect of hydraulic characteristics on the density or biomass of respective organisms (categorized into taxa or trophic guilds) [11–14]. Together, the models generate predictions on how discharge influences the habitat suitability for aquatic organisms (reviewed by [12,13]).

The relationship between hydraulic conditions and stream biota is influenced by the biological characteristics of the impacted species (e.g., life-stage requirements) and other abiotic pressures (e.g., water temperature and quality) [15,16]. Benthic macroinvertebrates have species-specific tolerance ranges for near-bed hydraulic conditions (hydraulic niche), with their occurrence and distribution responding to changes in physical habitat [17–19]. Hydraulic and substrate controls imposed by the physical habitat are thus master variables determining macroinvertebrate distributions [20] and the availability of resources [21,22]. From a metabolic perspective, macroinvertebrates' preferences for hydraulic habitat conditions can be seen as a tradeoff, where resistance to dislodgment in turbulent environments is balanced with foraging [23].

Macroinvertebrates can be assigned to functional feeding guilds (FFGs) defined by the combination of dominant feeding mechanisms and preferred resources [24]. The functional composition of the macroinvertebrate community in terms of resource use [25] can be applied to infer impacts of habitat degradation on ecological processes and energy pathways [26]. Hydromorphology also influences the distribution and retention of benthic organic matter in streams [27,28], and, in combination with nutrient availability, controls periphyton development [29]. Near-bed hydraulic conditions may thus, directly and indirectly, affect taxonomic and functional macroinvertebrate community composition and densities [30,31], i.e., through the effects on physical habitat and resource availability.

Several studies focused on physical habitat models and responses of groups of organisms, taxa, and community traits to flow-related physical habitat alterations. This led to a framework for streams, catchments, and regional predictions of the consequences of flow regulation [14,32–34]. However, although an exceedingly high number of alpine catchments is affected by flow regulation for hydropower [35], this biome is only marginally represented in such studies despite its unique hydro-morphological characteristics. As a consequence, habitat tolerances and responses to flow alteration could differ from those in other regions and biomes [14,36,37]

We collected high-resolution data on habitat, macroinvertebrate, and resource (particulate organic matter) distribution from two alpine streams to derive empirical preference models for invertebrate taxa, FFGs, and resource distribution. We coupled these preference models with a hydraulic model to predict how flow regulation of alpine streams may affect macroinvertebrate communities and their resources. Thereby, we aimed to develop and test tools that can assist flow management in alpine catchments to benefit macroinvertebrate communities that are increasingly under pressure from hydropower exploitation.

## 2. Materials and Methods

### 2.1. Study Sites

The study was carried out between December 2015 and May 2016 on the lower reaches of two 5th order alpine streams, Lesgiüna and Orino, in Switzerland, upstream of their confluence with the Brenno River (Figure 1). Since 1959, their catchments have been part of the OFIBLE hydropower network, which abstracts approximately one third of water from the catchments [38] and strongly influences the flow regimes of the streams [39]. The streams in this study have similar climatic and geological features. The Lesgiüna catchment covers an area of 36 km$^2$, of which 66% are above the water intakes. The Orino catchment has an area of 72 km$^2$, of which 86% are above the Malvaglia reservoir. There is no residual flow released from the reservoir and streamflow is restored only by tributaries downstream. Both catchments have minimal agricultural and residential influence, although the Orino flows in a slightly more anthropized landscape, which includes the small residential area of Malvaglia and some cultivated fields.

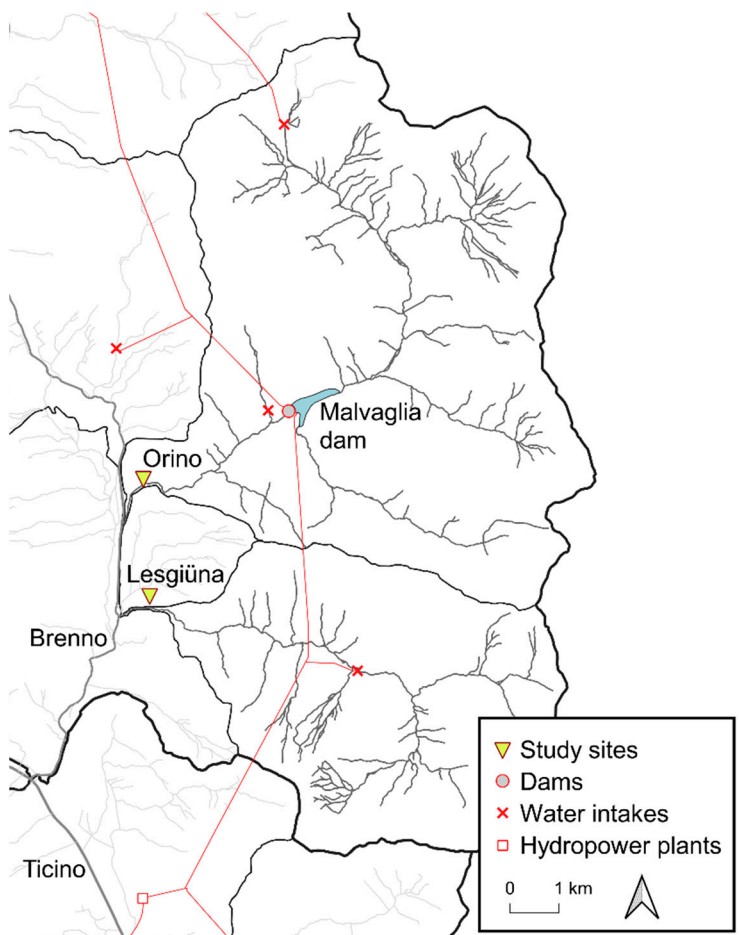

**Figure 1.** Overview of the study streams, catchments, and the hydropower network. Source of data: Swiss Federal Office of Topography.

For each stream, one ~70-m long reach was selected as study reach. The length of the study reaches was defined to obtain hydro-morphologically uniform reaches of sufficient length that provide enough evenly spaced replicate sampling spots needed for the habitat models (described below). The reach on the Lesgiüna (Figure 1) was a long run with a streambed mostly composed of medium-large cobbles (a-axis between 32 and 128 mm). The study site was located ~300 m below the exit from a narrow gorge, and ~200 m before entering the Brenno floodplain. This river section had a relatively homogeneous width (~9.5 m), laterally controlled by rock bank protections. The reach on the Orino (Figure 1)

had more complex morphology and featured two riffle-pool sequences. The study site was located ~200 m before the confluence with the Brenno river and ~200 m downstream from the last weir. The average channel width was 5 m; the streambed was similar to the Lesgiüna, although larger particles were more embedded (authors' observation).

### 2.2. Data Collection and Treatment

Physical habitat features (flow velocity, depth, sediment size, and FST) were measured along 17 transects (~4 m apart) perpendicular to flow at each site, at three discharges. Discharges differed roughly by a factor of 4: low winter discharge (baseflow), medium early spring discharge, and relatively high snowmelt discharge in late spring (respectively, Orino: 0.02, 0.09, 0.43 $m^3 \cdot s^{-1}$; Lesgiüna: 0.06, 0.26, 1.15 $m^3 \cdot s^{-1}$). Seventeen transects were the maximum feasible for sampling in one day with a team of three people. Doing so ensured that all measurements and samples were taken under constant discharge conditions. On each sampling day, discharge was calculated from measurements taken with a portable electromagnetic water flow meter (OTT MF pro, OTT, Kempten, Germany) at ~30 cm intervals, at three to four depths per point along the same cross-section. Cross-sections had a regular profile and were located immediately upstream of the study reaches. To maximize the accuracy of measurements, the streambed was smoothened by removing larger cobbles. The same instrument was used to measure depth and bottom flow velocity (30 s averages) at five evenly spaced measuring spots per transect, with the first and last spot of each transect ca. 25–30 cm from shore. Relative abundances of nine substrate size classes (<0.06; 0.06−2; 2−8; 8−16; 16−32; 32−64; 64−128; 128−256; >256 mm) were visually measured at each spot using a $20 \times 25$ cm grid and a weighted mean of sediment size was calculated (S1).

At each measuring spot along the transects, we used FST (FliesswasserStammTisch) hemispheres developed by Statzner and Müller [40] to estimate bottom shear stress [40,41], a key feature of near-bed hydraulic conditions. Shear stress controls abiotic characteristics of benthic habitats in streams and exerts selective pressure on flow-related traits of macroinvertebrates. Benthic macroinvertebrate distributions respond to FST range [14], indicating the suitability of FST hemispheres to assess flow-related macroinvertebrate habitat preferences. FST-hemispheres have identical shape and volume but a gradient in weight, due to differences in density of the material used [40]. They are thus labelled from no. 1 (the lightest) to no. 24 (the heaviest). Spot-scale shear stress was estimated from the number of the heaviest FST hemisphere that was not moved by flow. The operational procedure consists in horizontally inserting a base plate ($13 \times 18 \times 0.8$ cm) into the top layer of sediment. Hemispheres are then placed on the plate, one at a time, starting from the higher numbers and noting the number of the densest hemisphere that is moved by the force of the current. When even the lightest hemisphere was not moved, the measurement was ascribed to the fictional "no. 0" hemisphere. Accurate hemisphere placement was challenging with the original FST hemisphere setup [40]: hemispheres strongly adhered to the plate due to the smooth contact surfaces, thus impeding accurate measurements [42,43]. This effect was avoided by placing a paint-coated grid (3 mm wire, 1 cm mesh size) on the base plate to create a structured surface [44].

Sampling of benthic macroinvertebrates and particulate organic matter (fine POM (FPOM): 0.25−1 mm; coarse POM (CPOM): > 1 mm) was carried out in early spring 2016 during a period of stable discharge conditions. Sampling in this period minimized the risk of recent disturbance by flood events. In each reach, three replicate samples were taken from spots with FST numbers between 0 and 9 (three samples per number). Because FSTs were measured last in the sampling spot to minimize disturbance of the macroinvertebrate community and resources, we knew the exact FST number only after collecting the benthic sample. We thus completed the three replicates for each FST number 0 to 9 by visually selecting remaining sampling spots in the study reaches. Further samples were collected from spots with FST numbers up to no. 14 to improve preference-curve fitting (described below). This procedure resulted in 60 benthic samples (Lesgiüna (N = 32): three samples

for FST hemispheres from no. 0 to 7, two for no. 8, three for no. 9, one each for nos. 10, 12, 14; Orino (N = 28): three for nos. 0 to 8 and one for no. 12). Benthic samples were collected using a 25 × 25 cm Surber sampler (mesh size 0.25 mm). Sediment composition was noted, and flow velocity and depth were measured as for the physical habitat survey. In front of the Surber net, larger cobbles were brushed, and the substrate homogenously disturbed with a screwdriver to a depth of 14 cm for 60 s. Samples were stored in 70% ethanol. FST hemisphere numbers were measured after macroinvertebrate sampling because previous substrate disturbance does not seem to affect the measurement [40].

Water physico-chemistry data (Table 1; collected four times between February and May 2016) was obtained with a handheld multiparameter probe (HACH HQ40d, HACH, Loveland, CO, USA), and from water samples analyzed in the laboratory (AUA laboratory, Eawag, Switzerland).

**Table 1.** Water physico-chemistry of the study reaches between February and May 2016 (mean ± standard deviation, *N* = 4). DOC, dissolved organic carbon; $k_{20\,°C}$, conductivity at 20 °C; TN, total nitrogen; TP, total phosphorus; DO, dissolved oxygen; t, temperature.

| | DOC mg C/L | $k_{20\,°C}$ µs/cm$^2$ | pH | TN mg /L | PO$_4$–P µg /L | TP µg /L | DO [1] % | t [1] °C |
|---|---|---|---|---|---|---|---|---|
| Orino | 3.3 ± 0.4 | 68 ± 32 | 7.1 ± 0.5 | 1.1 ± 0.4 | 1.5 ± 0.6 | 5.7 ± 2.6 | 100.4 ± 1.4 | 6.4 ± 3.7 |
| Lesgiüna | 2.0 ± 0.5 | 46 ± 17 | 7.0 ± 0.0 | 0.8 ± 0.2 | 1.5 ± 0.2 | 3.2 ± 2.0 | 101.2 ± 2.3 | 4.7 ± 2.5 |

[1] Measured in the field.

In the laboratory, macroinvertebrates were separated from organic matter and sediment. Living roots were removed before CPOM and FPOM separation. Some samples from the Orino had a large amount of living root fragments (i.e., not part of particulate organic matter). These, as well as seeds and large wood fragments, were removed from all samples before drying (50 °C, 48 h) and incineration (550 °C, 3 h) to measure ash-free dry mass (AFDM).

Macroinvertebrates were identified to the lowest practical taxonomic level (i.e., mostly species or genus for EPTs and Coleoptera, family, or tribe for Diptera, and variable for the remaining taxa) using a dissecting microscope (M5, Wild Heerbrugg AG, Heerbrugg, Switzerland) and dichotomous identification keys [45–50]. Species-specific average individual dry-mass (DM) of macroinvertebrates was estimated from measurements (±0.05 mm) of head capsule width or body length (depending on the type of allometric equation available from the literature [51–55]) for a total of 40 individuals per taxa from each stream. Taxa for which allometric equations were unavailable accounted for less than 3% of the total abundance and were excluded from community biomass calculations. Dry-weight estimates were used to calculate total macroinvertebrate biomass per sample [24] and the relative biomass of dominant functional feeding groups (FFG) as follows: shredders, grazers or scrapers, passive filter feeders, detritus collector–gatherers, and predators. The relative FFG biomass was calculated by multiplying the biomass of a given taxon with an affinity score (1−10) as described in Moog [24]. In addition, the following indices were calculated per sample based on the relative abundance of macroinvertebrate taxa: taxa richness, Shannon (*H*), and Simpson (*D)* index:

$$H = -\sum \frac{n_i}{N} \cdot \ln \frac{n_i}{N} \tag{1}$$

$$D = 1 - \sum \frac{n_i(n_i - 1)}{N(N - 1)} \tag{2}$$

where *N* is the total number of taxa, and $n_i$ is the number of individuals of taxa *i*.

### 2.3. Data Analysis

Preference models were generated for a total of 14 taxa that met the following criteria: at least 100 individuals in total and occurrence in >10% of the samples at each site. Preference models were also fitted for FFGs, resources (CPOM, FPOM), and community metrics. The models chosen in this study were quadratic polynomial regressions [34]. Predicted values were standardized to obtain a Suitability Index (SI) [11] ranging from 0 (non-suitability) to 1 (maximum suitability or optimal habitat). Before fitting, response variables were log+1 transformed and physical habitat variables centered on the mean. These analyses were carried out using the statistical software R (version 3.6.0) [56].

To estimate discharge-dependent changes in near-bed hydraulic conditions in the two streams, we used the hydraulic model embedded in the FST-based CASiMiR module Benthos (Computer Aided Simulation Model for Instream Flow Requirements [57–59]). This model predicts the statistical distribution of FST numbers for the study reach under different discharge conditions expressed as the respective area of each FST number (m$^2$). The prediction is based on a density function of FST numbers that was calibrated with field measurements, by fitting the observations to a log-normal distribution.

The model also coupled the predicted FST distribution with the preference models (see above) to calculate the Weighted Usable Area (WUA) and the Hydraulic Habitat Suitability (HHS) for each response variable and each discharge [11].

$$WUA = \sum_{i=1}^{N} A_i \cdot HSI_i. \quad \left[ m^2 \right] \tag{3}$$

$$HHS = WUA / \sum_{i=1}^{N} A_i. \quad [-] \tag{4}$$

where $N$ is the total number of inundated spatial units (measuring spots) within the reach [-]; $A_i$ is the surface area of spatial unit $i$ [m$^2$]; and $HSI_i$ is the habitat suitability index of spatial unit $i$ [-].

The models allowed predicting taxonomic and functional community composition in response to discharge conditions. Preference models were fitted to data collected in spring, during intermediate discharge conditions (Table 2). The hydraulic models were fitted using data measured in winter (low discharge), early (medium discharge) and late spring (high discharge). The difference between HHS and WUA modelled at the low and mid discharge ($\Delta_1$), (Table 2) and the cumulative discharge increase $\Delta_1+\Delta_2$ ($\Delta_2$ being the difference between mid and high discharge) were calculated to highlight changes in HHS and WUA with changing discharge.

**Table 2.** Measured values of wetted area; FST$_\mu$, mean FST number approximated to the closest unit; V$_\mu$, mean velocity; D$_\mu$, mean depth (mean ± standard deviation); DGS, dominant grain size class under different discharge conditions (season).

| | Discharge (m$^3 \cdot$s$^{-1}$) | Wetted Area (m$^2$) | FST$_\mu$ (N) | V$_\mu$ (ms$^{-1}$) | D$_\mu$ (m) | DGS (mm) |
|---|---|---|---|---|---|---|
| **Orino** | | | | | | |
| Winter (baseflow) | 0.02 | 2848 | 1 ± 1.7 | 0.06 ± 0.07 | 0.06 ± 0.04 | 64–128 |
| Early spring | 0.09 | 3236 | 3 ± 2.4 | 0.11 ± 0.11 | 0.07 ± 0.05 | 64–128 |
| Late spring | 0.43 | 3596 | 5 ± 3.0 | 0.19 ± 0.15 | 0.15 ± 0.06 | 64–128 |
| **Lesgiüna** | | | | | | |
| Winter (baseflow) | 0.06 | 4567 | 1 ± 1.8 | 0.07 ± 0.06 | 0.07 ± 0.03 | 64–128 |
| Early spring | 0.26 | 6617 | 3 ± 2.4 | 0.16 ± 0.14 | 0.10 ± 0.10 | 64–128 |
| Late spring | 1.15 | 8885 | 5 ± 3.1 | 0.20 ± 0.18 | 0.15 ± 0.07 | 128–256 |

Measured in the field.

## 3. Results

### 3.1. Water Physico-Chemistry and Physical Habitat Assessment

Nitrogen, phosphorus, and dissolved organic carbon concentration levels were low and varied little between streams and among sampling dates (Table 1). Dissolved oxygen levels were always near saturation. Water temperature was on average two degrees higher on the Orino, where also a slightly higher fluctuation was observed.

FST mean number, water velocity, and depth varied consistently between the two streams at the three different discharges (Table 2). The dominant sediment grain size class was constant within sites except that it was higher in late spring at the Lesgiüna. The two smaller sediment size classes (<0.06 mm; 0.06−2 mm) were relatively more abundant in the Orino than the Lesgiüna (12% versus 2%). By contrast, Lesgiüna had substantially more boulders and large cobbles (36%) than Orino (11%).

### 3.2. Macroinvertebrates and Benthic Organic Matter

A total of 64 taxa were identified, of which 14 at the species level (Appendix C, Table A3). Macroinvertebrate communities were almost entirely composed of insects, with <0.01% belonging to other phyla (i.e., Annelida, Arachnida). Both sites had relatively high taxonomic diversity (Simpson index mean value of 0.72 ± 0.07 SD). The most abundant taxa were *Baetis alpinus* (total 31.8%), Orthocladiinae (25%), *Baetis rhodani* (17.6%) and *Leuctra* spp. (4.4%). Biomass was dominated by *B. alpinus* (40.6%), *B. rhodani* (22.2%), *Leuctra* spp. (7.8%) and Orthocladiinae (3.8%). Taxon richness ranged between 11 and 28, with no significant difference between sites, but was slightly higher in spots characterized by FST range 2–9. Mean biomass (ca. 9.0 DMg/m$^2$) and mean density (ca. 7600 ind/m$^2$) were similar between sites and peaked in the FST range 4–9. Grazers (44.2% of biomass) and collector-gatherers (41.4%) were the dominant FFGs, followed by predators (6.9%), shredders (4.3%), and passive filter feeders (3.2%). Notably, shredder biomass in the Lesgiüna was three times that of the Orino. In the Lesgiüna, mean CPOM biomass was 18.0 AFDMg/m$^2$ and mean FPOM 5 AFDMg/m$^2$. In the Orino, mean CPOM biomass was 8.0 AFDMg/m$^2$ and mean FPOM 7.5 AFDMg/m$^2$

### 3.3. Hydraulic Model

Figure 2 shows how the hydraulic model estimated FST frequencies at measured discharges and linearly interpolated these values to generate intermediate distributions. At both sites, increasing discharge reduced the area of 0 numbers, increased low-to-high numbers (1–10), and slightly increased very high numbers (11–16). FST numbers between 4 and 10 showed the greatest relative increase in surface, which means that, as expected, streambed shear stress increased overall with increasing discharge. Measured FST distribution during transect surveys, and a comparison between measured and modelled frequency distributions of FST numbers at both sites are reported in Appendix A (Figures A1 and A2).

### 3.4. Preference Model

The curves fitted using the preference models varied among the response variables. For some variables, the optimal suitability value ($SI_{max}$ = 1) was outside the range of modelled FST numbers (0−16). Three main patterns of responses were observed (Table 3): (a) a bell-shaped curve showing marked increasing/decreasing preference and an optimum peak (e.g., *B. rhodani*, *B. alpinus*); (b) a bowl-shaped curve, close to linear, with rapidly increasing suitability starting from ~0 and no detected optimum (e.g., *Protonemura* spp., *Hydropsyche* spp.); and (c) a relatively flat curve, with very high suitability values for all hemispheres (e.g., Orthocladiinae, FPOM). Model fit differed strongly among taxa, with best fits achieved for *E. sylvicola* ($R^2$ = 0.39), *B. alpinus* ($R^2$ = 0.48), *Protonemura* spp. ($R^2$ = 0.54), *Hydropsyche* spp. ($R^2$ = 0.40), *Rhyacophila* spp. ($R^2$ = 0.34), and Simuliidae ($R^2$ = 0.43).

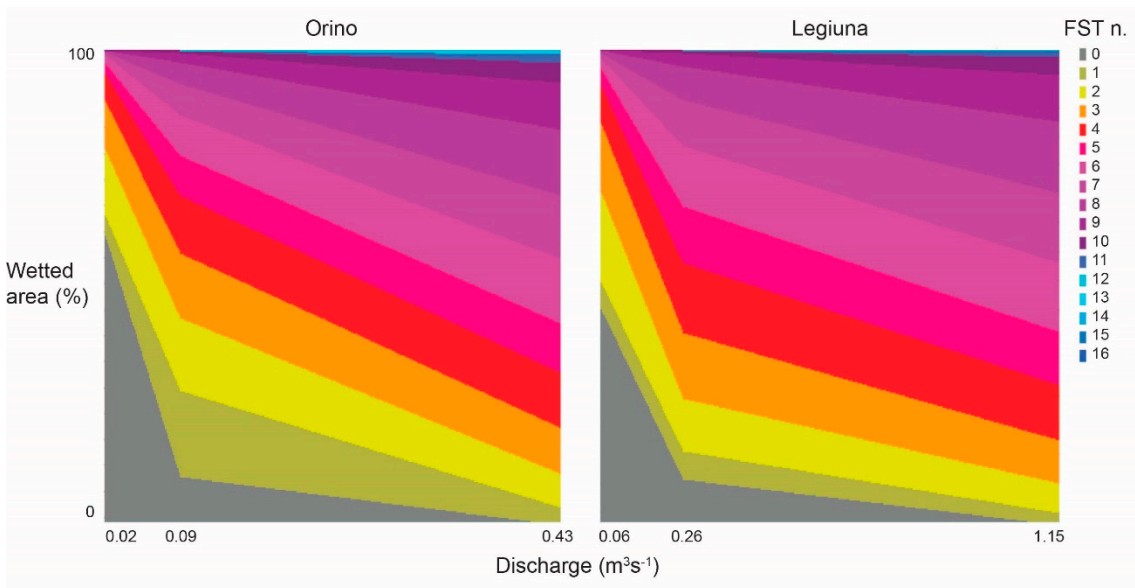

**Figure 2.** Relative wetted area for FST hemispheres at measured discharges, as modelled by CASiMiR module Benthos. X−axis scale differs between sites.

Models for taxa including *Alainites muticus*, Tanypodinae, *Leuctra* spp., Orthocladiinae, *Ecdyonurus* spp. and *Isoperla* spp. had a low fit ($R^2 < 0.2$). For the FFGs, models for grazers and passive filter feeders had the best fit, whereas model fit was poor for those of shredders, detritus collector-gatherers, and predators. Of the calculated community metrics, only total biomass and total abundance had $R^2 \geq 0.2$. Models for resources (CPOM and FPOM) performed poorly ($R^2 < 0.2$).

We combined the hydraulic habitat models with the preference models to estimate HHS and WUA for taxa, FFGs, organic resources, and community metrics at different discharge conditions (Appendix B, Tables A1 and A2). Based on these results, ΔHHS and ΔWUA were calculated to highlight changes with different discharges (Figure 3). Both sites showed similar response patterns. Because of differences in stream size, ΔWUA results varied in magnitude between the two sites. Most of the taxa showed an increase in HHS for both sites as discharge increased from low to intermediate (0.09 $m^3$ $s^{-1}$ for Orino and 0.26 $m^3$ $s^{-1}$ for Lesgiüna), except for *Ecdyonurus* spp. and Tanypodinae. With further increases in discharge, *B. rhodani* and *A. muticus* had decreasing HHS, while it remained constant (Orthocladiinae) or increased for other taxa. Rheophilic taxa such as *R. semicolorata*, *Rhithrogena* spp. and *Isoperla* spp. showed the largest positive ΔHHS with discharge increase. For FFGs, passive filter feeders showed the strongest response in terms of ΔHHS (Figure 3). For other FFGs, the cumulative increase in HHS from low to high discharge (i.e., $\Delta_1 + \Delta_2$) remained below 0.1. Similar results were found for community metrics and CPOM, while FPOM was the only variable showing a decrease in HHS for both discharge increments (i.e., $\Delta_1$ and $\Delta_1 + \Delta_2$.).

**Table 3.** $R^2$ values of regressions and suitability indices of FSTs (range 0–16). Variables marked with + have SImax < 1. ***: $p < 0.001$, **: $p < 0.01$, *: $p < 0.05$.

| | $R^2$ | FST 0 | 1 | 2 | 3 | 4 | 5 | 6 | 7 | 8 | 9 | 10 | 11 | 12 | 13 | 14 | 15 | 16 |
|---|---|---|---|---|---|---|---|---|---|---|---|---|---|---|---|---|---|---|---|
| **Taxa** | | | | | | | | | | | | | | | | | | | |
| *E. sylvicola* +*** | 0.39 | 0 | 0.03 | 0.07 | 0.1 | 0.13 | 0.16 | 0.20 | 0.24 | 0.27 | 0.31 | 0.35 | 0.39 | 0.43 | 0.47 | 0.52 | 0.56 | 0.6 |
| *Ecdyonurus* spp.* | 0.10 | 1 | 0.98 | 0.95 | 0.92 | 0.88 | 0.84 | 0.79 | 0.73 | 0.67 | 0.60 | 0.53 | 0.44 | 0.36 | 0.27 | 0.17 | 0.06 | 0.05 |
| *R.semicolorata* *** | 0.29 | 0.01 | 0.15 | 0.30 | 0.43 | 0.55 | 0.66 | 0.75 | 0.82 | 0.89 | 0.94 | 0.97 | 0.99 | 1 | 0.99 | 0.97 | 0.94 | 0.89 |
| *B. alpinus* *** | 0.48 | 0.42 | 0.55 | 0.66 | 0.76 | 0.84 | 0.90 | 0.95 | 0.98 | 1 | 0.99 | 0.98 | 0.96 | 0.91 | 0.85 | 0.77 | 0.68 | 0.57 |
| *B. rhodani* ** | 0.20 | 0.84 | 0.91 | 0.95 | 0.99 | 1 | 0.99 | 0.98 | 0.94 | 0.89 | 0.82 | 0.73 | 0.062 | 0.50 | 0.36 | 0.20 | 0.03 | 0 |
| *A.muticus* | 0.04 | 0.82 | 0.90 | 0.95 | 0.99 | 1 | 0.99 | 0.97 | 0.92 | 0.86 | 0.77 | 0.66 | 0.54 | 0.39 | 0.22 | 0.04 | 0 | 0 |
| *Leuctra* spp. +* | 0.12 | 0.36 | 0.37 | 0.39 | 0.41 | 0.42 | 0.44 | 0.46 | 0.48 | 0.51 | 0.53 | 0.55 | 0.58 | 0.60 | 0.63 | 0.66 | 0.69 | 0.72 |
| *Isoperla* spp. ** | 0.17 | 0.30 | 0.41 | 0.52 | 0.61 | 0.70 | 0.77 | 0.84 | 0.89 | 0.93 | 0.96 | 0.99 | 1 | 0.99 | 0.98 | 0.97 | 0.94 | 0.90 |
| *Protonemura* spp. +*** | 0.54 | 0 | 0.02 | 0.03 | 0.05 | 0.07 | 0.09 | 0.12 | 0.14 | 0.17 | 0.21 | 0.24 | 0.28 | 0.32 | 0.36 | 0.41 | 0.45 | 0.50 |
| *Hydropsyche* spp. +*** | 0.40 | 0.08 | 0.12 | 0.15 | 0.19 | 0.23 | 0.26 | 0.30 | 0.34 | 0.37 | 0.41 | 0.45 | 0.49 | 0.52 | 0.56 | 0.60 | 0.64 | 0.68 |
| *Rhyacophila* spp.*** | 0.34 | 0.10 | 0.22 | 0.33 | 0.44 | 0.53 | 0.62 | 0.69 | 0.76 | 0.82 | 0.87 | 0.92 | 0.95 | 0.98 | 0.99 | 1 | 0.99 | 0.98 |
| Simuliidae +*** | 0.43 | 0.01 | 0.05 | 0.09 | 0.13 | 0.17 | 0.21 | 0.25 | 0.29 | 0.33 | 0.38 | 0.42 | 0.46 | 0.50 | 0.54 | 0.58 | 0.62 | 0.66 |
| Tanypodinae | 0.10 | 1 | 0.98 | 0.96 | 0.92 | 0.88 | 0.82 | 0.76 | 0.68 | 0.60 | 0.50 | 0.39 | 0.28 | 0.15 | 0.01 | 0 | 0 | 0 |
| Orthocladiinae | 0.01 | 0.95 | 0.96 | 0.97 | 0.98 | 0.99 | 0.99 | 1 | 1 | 1 | 1 | 1 | 0.99 | 0.99 | 0.98 | 0.97 | 0.96 | 0.94 |
| | | | | | | | | | | | | | | | | | | |
| **FFGs** | | | | | | | | | | | | | | | | | | | |
| Grazers *** | 0.33 | 0.75 | 0.80 | 0.84 | 0.88 | 0.92 | 0.94 | 0.97 | 0.98 | 0.99 | 1 | 1 | 0.99 | 0.98 | 0.97 | 0.95 | 0.92 | 0.89 |
| Shredders +** | 0.20 | 0.41 | 0.43 | 0.45 | 0.46 | 0.48 | 0.50 | 0.52 | 0.54 | 0.56 | 0.58 | 0.60 | 0.63 | 0.65 | 0.68 | 0.70 | 0.73 | 0.76 |
| P. filter feeders +*** | 0.46 | 0.13 | 0.19 | 0.24 | 0.30 | 0.35 | 0.40 | 0.45 | 0.49 | 0.54 | 0.58 | 0.62 | 0.66 | 0.70 | 0.73 | 0.77 | 0.80 | 0.83 |
| Detritus C.G. ** | 0.17 | 0.85 | 0.89 | 0.92 | 0.94 | 0.96 | 0.98 | 0.99 | 1 | 1 | 1 | 0.99 | 0.98 | 0.96 | 0.94 | 0.92 | 0.89 | 0.85 |
| Predators ** | 0.14 | 0.68 | 0.71 | 0.74 | 0.76 | 0.79 | 0.81 | 0.84 | 0.86 | 0.88 | 0.90 | 0.91 | 0.93 | 0.94 | 0.96 | 0.97 | 0.98 | 1 |
| **Comm. metrics** | | | | | | | | | | | | | | | | | | | |
| Tot. biomass *** | 0.26 | 0.83 | 0.86 | 0.89 | 0.91 | 0.94 | 0.95 | 0.97 | 0.98 | 0.99 | 1 | 1 | 1 | 1 | 0.99 | 0.98 | 0.96 | 0.95 |
| Tot. abundance ** | 0.20 | 0.87 | 0.90 | 0.93 | 0.95 | 0.97 | 0.98 | 0.99 | 1 | 1 | 1 | 1 | 0.99 | 0.98 | 0.96 | 0.94 | 0.92 | 0.89 |
| Taxa richness * | 0.16 | 0.90 | 0.91 | 0.92 | 0.94 | 0.95 | 0.96 | 0.97 | 0.98 | 0.99 | 1 | 1 | 1 | 1 | 1 | 0.99 | 0.99 | 0.98 |
| Shannon index * | 0.16 | 0.70 | 0.70 | 0.70 | 0.71 | 0.71 | 0.72 | 0.73 | 0.73 | 0.74 | 0.75 | 0.76 | 0.77 | 0.79 | 0.80 | 0.81 | 0.83 | 0.84 |
| Simpson index * | 0.12 | 0.82 | 0.83 | 0.84 | 0.85 | 0.85 | 0.86 | 0.87 | 0.87 | 0.88 | 0.89 | 0.90 | 0.90 | 0.91 | 0.92 | 0.93 | 0.94 | 0.95 |
| **Resources** | | | | | | | | | | | | | | | | | | | |
| CPOM * | 0.18 | 0.10 | 0.10 | 0.10 | 0.11 | 0.11 | 0.12 | 0.14 | 0.16 | 0.18 | 0.20 | 0.23 | 0.26 | 0.30 | 0.34 | 0.38 | 0.43 | 0.48 |
| FPOM | <0.01 | 1 | 0.99 | 0.98 | 0.97 | 0.97 | 0.96 | 0.95 | 0.94 | 0.94 | 0.93 | 0.92 | 0.91 | 0.90 | 0.89 | 0.88 | 0.87 | 0.86 |

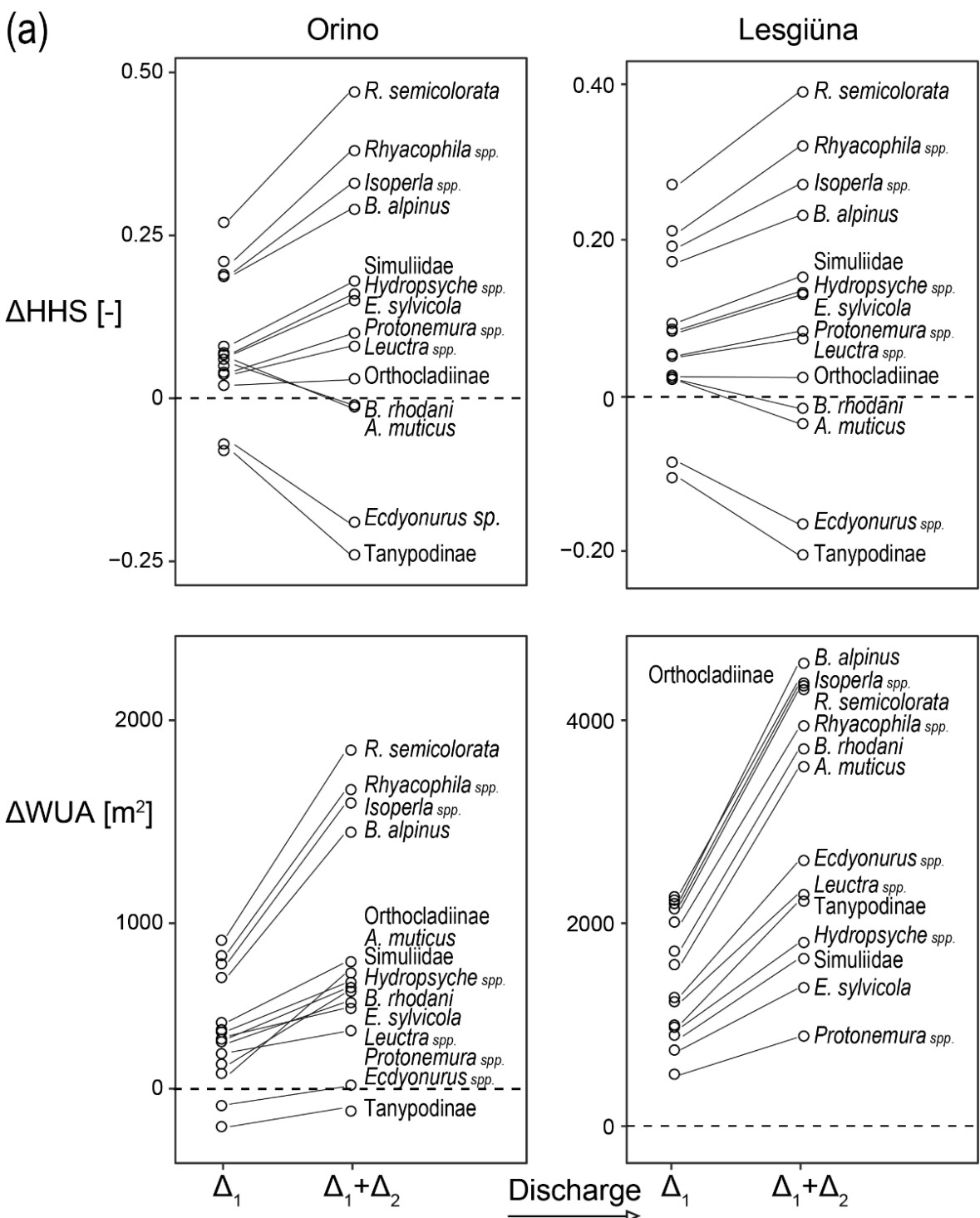

**Figure 3.** *Cont.*

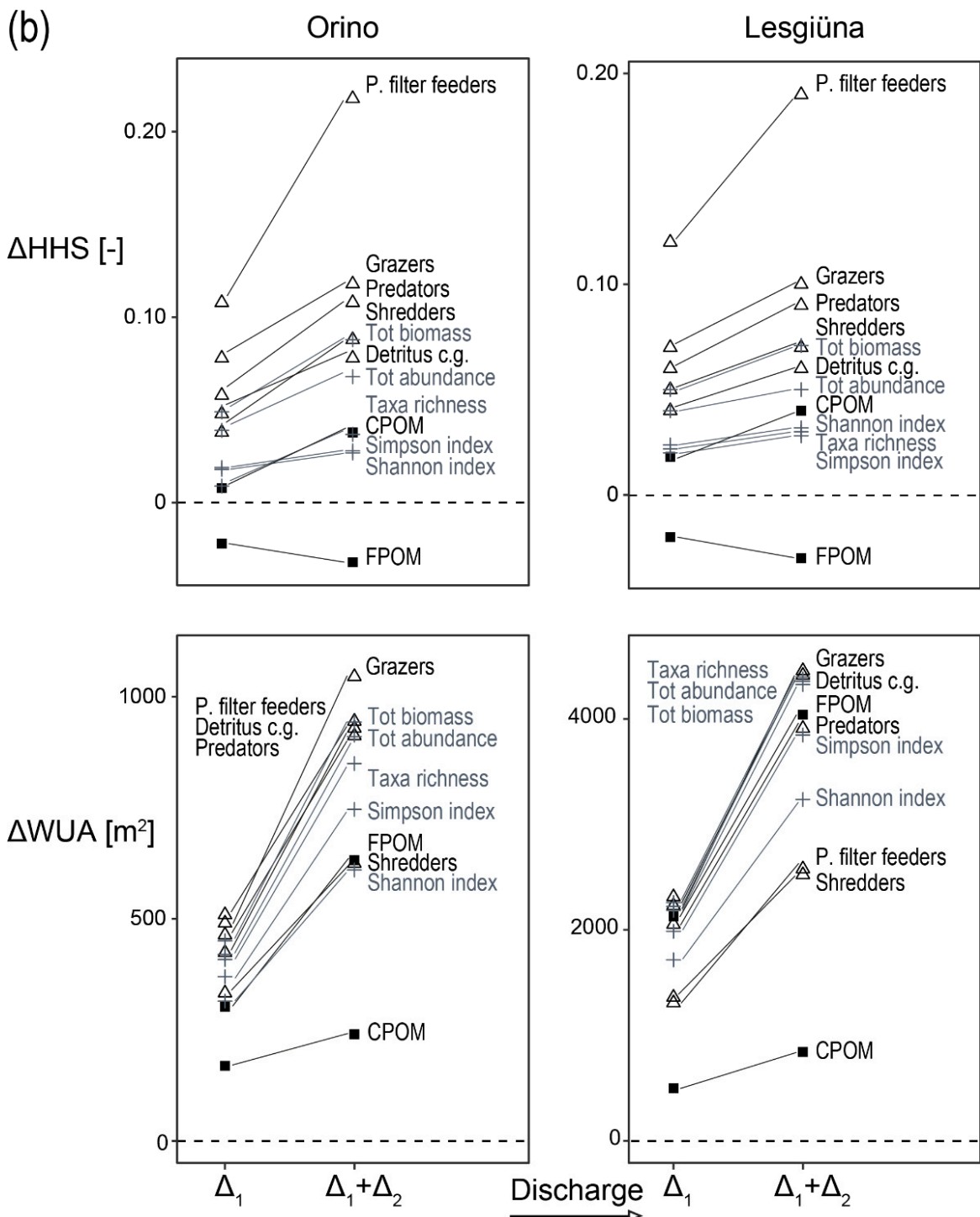

**Figure 3.** Changes in HHS and WUA with increasing discharge from low- to mid-flow ($\Delta_1$) and low to high flow ($\Delta_1+\Delta_2$), respectively. (**Panel a**): taxa; (**Panel b**): FFGs ($\Delta$), community metrics (+, in grey), and organic resources (■). $\Delta_1 > 0$ (dashed line) indicates that an increase in discharge from low to mid has positive effects on HHS or WUA, $\Delta_1 < 0$ indicates a negative effect of the discharge increase. The slope of the line connecting $\Delta_1$ with $\Delta_1+\Delta_2$ reveals whether a further increase in discharge has positive, neutral, or negative effects on HHS or WUA compared to $\Delta_1$ alone. Positive slopes mean that further increase in discharge from mid- to high-flows further increased HHS or WUA. Negative slopes mean that the further discharge increase reduced the positive effects of $\Delta_1$ (if $\Delta_1 > 0$; e.g., *B. rhodani* and *A. muticus*) or intensified the negative effects of $\Delta_1$ (if $\Delta_1 < 0$; e.g., *Ecdyonurus* sp. and Tanypodinae). Zero slopes mean that $\Delta_2$ does not affect HHS or WUA. Y-axis not in scale among panels.

## 4. Discussion

### 4.1. Hydraulic Models

The physical habitat models showed a predictable decrease in high-shear stress area under low discharge conditions (Figure 2), with a shift towards very low FST-hemisphere numbers at the lowest discharge (up to 50% of spots had the lowest measurable FST number). However, shear-stress distribution (measured as FST) did not always uniformly change with discharge (Appendix A, Figure A1). At the Lesgiüna, frequencies of FST numbers showed a bimodal distribution, where the first maximum remained constant at 0 and the second moved gradually from 2–4 to 6–8 with increasing discharge. At the Orino, the FST hemispheres frequency curve peaked at 0 at the lowest discharge, was inversely proportional to FST number at the moderate discharge and resembled that of the Lesgiüna at the highest discharge. Differences in FST frequency distribution with discharge could be attributed to differences in channel morphology [60,61].

Due to gentle lateral slopes of the Lesgiüna reach, areas with minimal shear stress (FST numbers 0) were maintained on the newly inundated margins when discharge increased. In contrast, the morphology of the Orino reach was characterized by a pronounced lateral discontinuity, owing to the presence of higher vegetated banks, dry secondary channels, and small, vegetated gravel bars. This morphology was possibly due to channel incision caused by limited sediment supply below the dam. Consequently, the banks were not inundated at intermediate discharge as the flow was constrained in the main channel. This led to a loss of low-flow habitats close to the banks with increasing discharge. A return to a bimodal distribution (including some 0 FST values) at the highest measured discharge follows the re-occurrence of low-flow habitat, as some areas on the banks became inundated (data not shown).

Discharge was also related to the overall hydraulic heterogeneity and the availability of key habitats of macroinvertebrates. At very low discharge (winter baseflow), high FST values (nos. 12–16) were rare in both streams. Nevertheless, the few spots with high shear stress present during baseflow (Figure 2) may be crucial to maintain rheophilic taxa and long-lived taxa whose habitat preference varies with life stage [31] (e.g., *Perla grandis* [62]). In comparison, at high discharge, flow refugia created by large wood debris and (in our study) boulders are crucial to protect macroinvertebrates from being dislodged [63,64].

### 4.2. Preference Models

Generally, the modelled hydraulic preferences of taxa covered in this study were consistent with published preference models [14]. Overall, spots with intermediate shear stress (FST range 4–11) were optimal for most taxa (Table 3). However, the predictive accuracy of the preference models was low to moderate. The variance not explained by the models can have several sources, including species-specific differences in habitat preference, unmeasured abiotic and biotic factors, tolerance to a range of measured habitat conditions, and stochastic distributions. In our study, coarse taxonomic identification might explain the apparent lack of hydraulic preferences for some taxa (e.g., Orthocladiinae: $R^2 = 0.01$), which confirms the large variation in hydraulic preferences in this taxon [65]. In addition to physical habitat conditions, other ecological mechanisms define the distribution of taxa within stream reaches, including different habitat requirements of different developmental stages (larger individuals generally occupying spots with higher bottom hydraulic forces [66]), colonization history [67], and biotic interactions (e.g., presence of predators, competitors, and resources) [68].

Furthermore, the measured shear stress distribution was in the range of $0-14$ FST numbers, thus not covering the upper portion of the FST range (the complete hemisphere set covers numbers 1 to 24). Although hydraulic preference generally declines at very high shear stress [14,32], the inclusion of spots with higher hemisphere numbers would have been important to increase the accuracy of the biological models, i.e., to capture decreasing habitat suitability with increasing shear stress for rheophilic taxa.

Preference models for FFGs had low predictive power. Nevertheless, our results suggest a preference of passive filter feeders for increasing shear stress (estimated as FST; $R^2$ = 0.46), probably as a consequence of their feeding mode and ability to resist relatively high shear stress [32,69]. Filter feeders in the study reaches were mostly represented by *Simulium*, *Prosimulium* and *Hydropsyche* sp., which strongly depend on flow for feeding. Conversely, many taxa had a variable grade of affinity to grazing, resulting in a moderate fit, but a relatively flat curve for this FFG, a consequence of the integration across taxa with variable flow-dependent habitat preferences.

### 4.3. Changes in Habitat Suitability in Response to Discharge

Based on our models, water abstraction (e.g., due to hydropower production or climate change) would result in less-suitable hydraulic conditions for most taxa, except *Ecdyonurus* spp., and Tanypodinae (Figure 3). Conversely, increased baseflow would improve habitat suitability for most taxa, although some, such as *B. rhodani* and *A. muticus*, would suffer a slight decrease in habitat suitability at high discharge. Nevertheless, the effect of an increased wetted area with increasing discharge would in most cases compensate for reductions in HHS (reflected in WUA). Exceptions include *Ecdyonurus* spp. and Tanypodinae in the Orino, which prefer low discharge conditions that would not be compensated by the increase in wetted area at higher discharge in that stream (Figure 3). The effect of increased discharge on habitat suitability was not linear. For most taxa, a change from low to medium discharge (winter to spring conditions) was predicted to have a greater beneficial effect on HHS than the increase in discharge from mid to high discharge (summer conditions in our study; Figure 3, Tables A1 and A2). This suggests that some factors (e.g., decreased suitability for taxa preferring intermediate FST) may attenuate the overall positive effects of further increasing discharge, but it also highlights the paramount importance of baseflow and residual flow management for macroinvertebrate communities in alpine streams.

### 4.4. Differences between Streams

In spring, the composition of macroinvertebrate communities was similar between both study sites, i.e., dominated by Baetidae and Chironomidae. However, relative abundances of respective taxa showed marked differences. As water physico-chemistry revealed no signs of anthropogenic impacts at both sites, these differences were probably mainly caused by hydro-morphological dissimilarities. The study reaches had similar average near-bed hydraulic conditions (measured as frequencies of FST numbers) during our study period despite substantial differences in discharge (Table 2). However, the frequency of spots with low shear stress was higher in the Orino. As a result, specialists for slow-flow areas (i.e., limnephilic taxa) with fine sediment (e.g., *Ephemera danica*, Odonata and Oligochaeta) were found almost solely in the Orino. In contrast, rheophilic taxa with preference for higher flow velocities and coarser substrate (e.g., *E. alpicola*, *Rhithrogena* spp.) were more abundant in the Lesgiüna.

The more natural flow regime in the Lesgiüna, with seasonal discharge variability and the associated regular streambed disturbance, probably selects for taxa with specific life history traits that allow them to benefit from floods and low flows between floods (see [70]). On the contrary, flow regulation and overall more stable conditions at the Orino would, in the long term, favour taxa which benefit from low-disturbance conditions [71,72]. Community composition and resource availability is also driven by variability in habitat characteristics on longer timescales. Hydrological variability of the stream might be particularly relevant for alpine streams that naturally have high variability but are often constrained by water abstraction and damming. The dam regulating the Orino reduced its hydrological variability and the frequency of floods with consequences on habitat conditions at the study reach, especially when compared to Lesgiüna, as shown in the occurrence of fine sediment cover and vegetated patches particularly in the Orino. The long-term (seasonal) effects of reduced discharge variability in the Orino were evident in

the presence of fine sediment patches and the occurrence of macroinvertebrate taxa that prefer these patches, despite similar flow conditions in both streams during the study.

### 4.5. Management Implications and Perspectives

Alpine river management in Europe is subject to two main bodies of law, the Water Framework Directive (WFD) [73] and the Swiss Water Protection Act (814.20). The former determines residual flows as a function of minimum flows ($Q_{347}$), without reference to ecological conditions. Good ecological status of rivers as defined in the WFD is determined by the assessment of the status of some biotic indicators, including macroinvertebrates. However, despite the well-acknowledged influence of flow regime on ecological status, the hydrological condition is solely intended as a supporting element of these biotic indicators [74]. Our results highlight two main findings that should be considered when assessing the status of macroinvertebrate communities based on flow-dependent habitat conditions and predicting the effect of flow alteration/restoration. We observed that: (1) a decline in baseflow below a certain stream-specific value can substantially alter the composition of the community, and (2) discharge-dependent thresholds could determine optimal conditions for macroinvertebrate taxa. These findings, merging biotic with abiotic conditions, could support the definition of ecologically relevant residual flow targets in streams where hydropower utilization is the dominant impact.

Taxonomic resolution influences the predictive accuracy of preference models. For instance, the large functional diversity within the subfamily Orthocladiinae could explain the poor fit of its preference model. Life history and seasonality probably further contribute to low predictive accuracy. Preference models based on sampling in a specific season are not representative for all larval stages, in particular if only one stage is present during sampling ("survey effect" [14]). In our study, the univoltine *Ecdyonurus* spp. was present as early instars (head capsule width $1.38 \pm 0.88$ mm), which were sensitive to strong hydraulic bottom forces (Table 3). This contrasts with its characterization as a rheo- to limnophilic taxa based on other larval stages [75], and with published hydraulic preference models [14,76]. This issue might be less relevant for multivoltine species, where the presence of several cohorts during sampling might result in preference models that are more representative for the species. However, it shows the value of study-specific preference models that can then be compared to published ones for interpretation. Seasonal differences are likely to play a role also for community-level and functional responses. For instance, Merigoux and Dolédec [32] observed that taxa richness was negatively correlated with FST numbers in spring and vice-versa in autumn. Seasonal differences in ecological conditions can modify the intensity of biotic interactions (in addition to the occurrence of different larval stages discussed above), and in turn, alter physical habitat preferences of taxa.

Applying the FFG concept to the same preference models based on flow-dependent physical habitat conditions resulted in high model fits only when feeding strategy was strongly related to flow (i.e., for passive filter feeders; $R^2 = 0.46$). Models were less accurate for other FFG, probably because of weak direct relationships between the distribution of their respective resources and near-bed hydraulic conditions (the basis of our hypothesis between FFG and flow conditions), as found in this and other studies [21,28,77]. This conclusion is supported by the distribution of shredders, which was strongly related to that of CPOM, whereas neither had a clear relationship with near-bed hydraulic conditions.

Evidently, generalization of our results requires further spatial and temporal validation. For instance, higher altitude portions of the catchments should be considered to cover a more representative gradient of conditions of alpine streams. Survey size, i.e., the range of hydraulic conditions and number of sites assessed, also strongly influences the definition of optimal habitat conditions of taxa and its precision [32,78–80]. Developing representative physical habitat models and preference curves for different stream types [60,61,81] would probably facilitate their generalization while also reducing the amount of field data required.

Community composition and relative abundances of taxa are influenced by present environmental conditions, including the effects of anthropogenic stressors such as water abstraction and climate change. Community adaptation to new conditions is a non-linear, multi-faceted ecological response involving several levels of change associated with the physical habitat that influences macroinvertebrate communities but also indirect effects through changes in resource availability and biotic interactions [82,83]. Indeed, it is challenging to predict long-term community shifts and ecosystem adjustments when a stressor is removed or added to an ecosystem (e.g., baseflow restoration or increase in water abstraction). We found near-bed hydraulic preference of macroinvertebrate taxa that are consistent with those reported from other studies. However, further development and validation of habitat models for alpine catchments and increasing seasonal and altitudinal range will be necessary to better describe hydraulic preferences of macroinvertebrate taxa and the distribution of their resources in alpine streams. This is needed to support policy makers, environmental managers, and hydropower operators in predicting the impacts of water withdrawal on taxonomic and functional macroinvertebrate community composition, as well as the ecosystem processes and services that depend on it. Our results point towards the existence of flow-related thresholds for the optimal near-bed hydraulic conditions to safeguard diverse and well-functioning alpine macroinvertebrate communities. However, adaptive water management should not only focus on base-flow levels but also draw on the recent advances in environmental flow design. These account for the natural flow regime (e.g., seasonal flow variability [84,85]) to sustain hydrogeomorphic processes that control ecological dynamics in rivers [86–88].

**Author Contributions:** Conceptualization, G.C., A.B., F.L., C.T.R.; methodology, G.C., A.B., F.L.; formal analysis, G.C.; data curation, G.C.; writing—original draft preparation, G.C., A.B.; writing—review and editing, G.C., A.B., F.L., C.T.R.; supervision, A.B., F.L., C.T.R. All authors have read and agreed to the published version of the manuscript.

**Funding:** This research was funded by the WACOMA Erasmus Mundus Scholarship.

**Data Availability Statement:** The data presented in this study are available on request from the corresponding author.

**Acknowledgments:** Thanks to the Institute of Earth Sciences of SUPSI for logistics and research material; the AUA laboratory at Eawag for water chemistry analyses; Mattia Domenici, Stefano Rioggi, Simona Bronzini, Filippo Franchini and Giorgia Camperio for their help during fieldwork; Maurizio Pozzoni, Sebastian Pera and Christian Scapozza for advice and inputs; Marco Simona for help in the laboratory; M. Forcellini (INRAE Lyon-Villeurbane) for the FSTs; and two anonymous reviewers for their constructive comments.

**Conflicts of Interest:** The authors declare no conflict of interest.

**Appendix A**

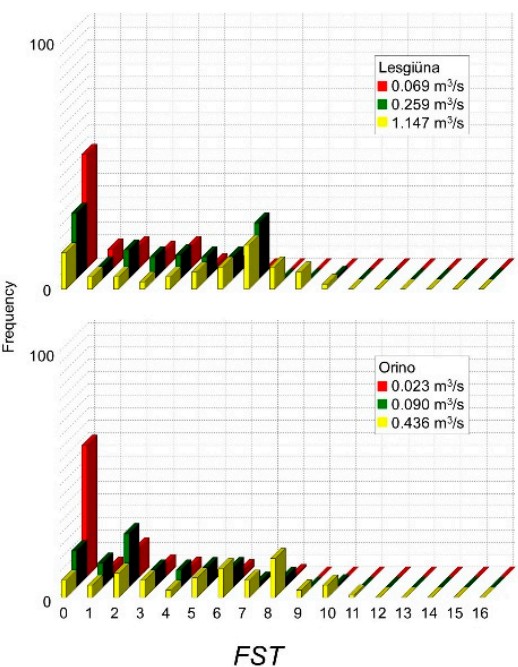

**Figure A1.** Area of measured FST hemispheres at different measured discharges in the Orino and the Lesgiüna.

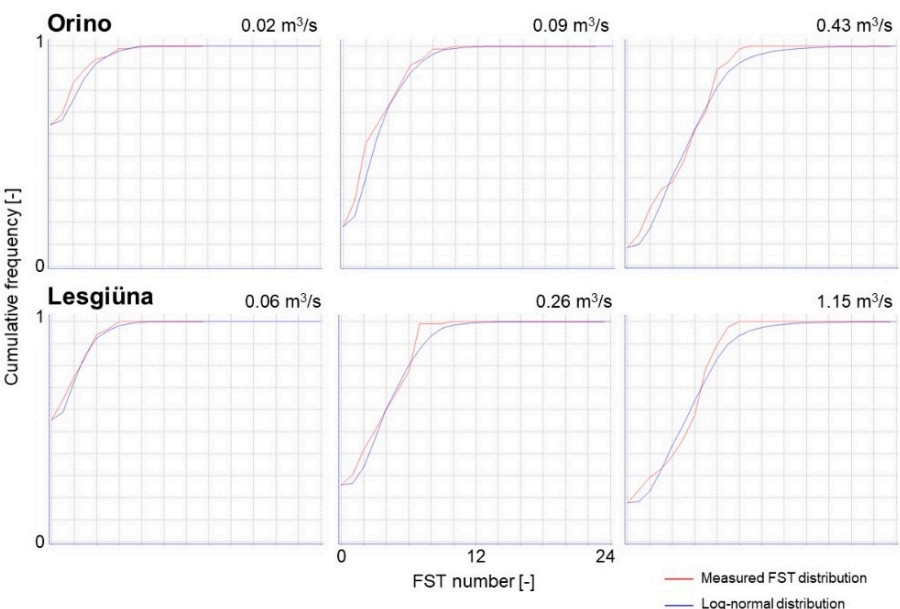

**Figure A2.** CASiMiR hydraulic model fit to observed values of FST numbers at the measured discharges in the Orino and the Lesgiüna.

## Appendix B

**Table A1.** Modelled WUA and HHS for the Orino.

| Discharge | WUA [m²] | | | HHS | | |
|---|---|---|---|---|---|---|
| | 0.02 m³ s⁻¹ | 0.09 m³ s⁻¹ | 0.44 m³ s⁻¹ | 0.02 m³ s⁻¹ | 0.09 m³ s⁻¹ | 0.44 m³ s⁻¹ |
| **Taxa** | | | | | | |
| *Baetis alpinus* | 1559.90 | 2379.01 | 3017.02 | 0.55 | 0.74 | 0.84 |
| *Baetis rhodani* | 2525.79 | 3035.85 | 3176.58 | 0.89 | 0.94 | 0.88 |
| *Baetis muticus* | 2418.04 | 3003.01 | 3081.16 | 0.87 | 0.93 | 0.86 |
| *Epeorus sylvicola* | 118.77 | 361.48 | 694.93 | 0.04 | 0.11 | 0.19 |
| *Rhitrogena semicolorata* | 463.54 | 1386.96 | 2272.84 | 0.16 | 0.43 | 0.63 |
| *Ecdyonurus* spp. | 2738.47 | 2874.96 | 2765.21 | 0.96 | 0.89 | 0.77 |
| *Isoperla* spp. | 1202.48 | 1973.67 | 2702.45 | 0.42 | 0.61 | 0.75 |
| *Leuctra spp.* | 1082.58 | 1343.66 | 1659.67 | 0.38 | 0.42 | 0.46 |
| *Protonemura* spp. | 63.63 | 203.62 | 439.97 | 0.02 | 0.06 | 0.12 |
| *Hydropsyche* spp. | 358.25 | 654.61 | 1035.15 | 0.13 | 0.20 | 0.29 |
| *Rhyacophila* spp. | 663.79 | 1426.78 | 2210.17 | 0.23 | 0.44 | 0.61 |
| Orthocladiinae | 2737.82 | 3164.39 | 3546.89 | 0.96 | 0.98 | 0.99 |
| Tanypodinae | 2732.76 | 2836.49 | 2586.80 | 0.96 | 0.88 | 0.72 |
| Simuliidae | 171.24 | 465.64 | 856.20 | 0.06 | 0.14 | 0.24 |
| **FFGs** | | | | | | |
| Grazers | 2279.42 | 2834.50 | 3324.77 | 0.80 | 0.88 | 0.92 |
| Shredders | 1232.08 | 1523.85 | 1856.78 | 0.43 | 0.47 | 0.52 |
| P. filter feeders | 564.83 | 1000.55 | 1509.66 | 0.20 | 0.31 | 0.42 |
| Detritus C.G. | 2519.10 | 3024.19 | 3448.24 | 0.88 | 0.93 | 0.96 |
| Predators | 2033.22 | 2487.05 | 2950.82 | 0.71 | 0.77 | 0.82 |
| **Comm. metrics** | | | | | | |
| Tot. abundance | 2563.62 | 3054.06 | 3473.86 | 0.90 | 0.94 | 0.97 |
| Tot. biomass | 2455.64 | 2948.32 | 3398.62 | 0.86 | 0.91 | 0.95 |
| Taxa richness | 2636.61 | 3074.33 | 3482.67 | 0.93 | 0.95 | 0.97 |
| Shannon index | 2004.65 | 2304.48 | 2619.02 | 0.70 | 0.71 | 0.73 |
| Simpson index | 2366.18 | 2743.91 | 3109.11 | 0.83 | 0.85 | 0.86 |
| **Resources** | | | | | | |
| CPOM | 317.33 | 388.42 | 557.32 | 0.11 | 0.12 | 0.15 |
| FPOM | 2817.32 | 3147.32 | 3449.35 | 0.99 | 0.97 | 0.96 |

**Table A2.** Modelled WUA and HHS for the Lesgiüna.

| Discharge | WUA [m²] | | | HHS | | |
|---|---|---|---|---|---|---|
| | 0.07 m³ s⁻¹ | 0.26 m³ s⁻¹ | 1.15 m³ s⁻¹ | 0.07 m³ s⁻¹ | 0.26 m³ s⁻¹ | 1.15 m³ s⁻¹ |
| **Taxa** | | | | | | |
| *Baetis alpinus* | 2636.38 | 4970.42 | 7198.79 | 0.58 | 0.75 | 0.81 |
| *Baetis rhodani* | 4102.96 | 6096.36 | 7821.66 | 0.90 | 0.92 | 0.88 |
| *Baetis muticus* | 4044.52 | 5996.58 | 7588.22 | 0.89 | 0.91 | 0.85 |
| *Epeorus sylvicola* | 233.07 | 848.19 | 1598.28 | 0.05 | 0.13 | 0.18 |
| *Rhitrogena semicolorata* | 923.63 | 3084.90 | 5226.90 | 0.20 | 0.47 | 0.59 |
| *Ecdyonurus* spp. | 4353.69 | 5704.12 | 6972.5 | 0.95 | 0.86 | 0.78 |
| *Isoperla* spp. | 2055.81 | 4209.87 | 6404.58 | 0.45 | 0.64 | 0.72 |
| *Leuctra* spp. | 1756.25 | 2812.96 | 4039.43 | 0.38 | 0.43 | 0.45 |
| *Protonemura* spp. | 124.33 | 500.14 | 1011.87 | 0.03 | 0.08 | 0.11 |
| *Hydropsyche* spp. | 621.15 | 1456.32 | 2431.77 | 0.14 | 0.22 | 0.27 |
| *Rhyacophila* spp. | 1202.68 | 3136.64 | 5148.59 | 0.26 | 0.47 | 0.58 |
| Orthocladiinae | 4402.18 | 6480.70 | 8742.53 | 0.96 | 0.98 | 0.98 |
| Tanypodinae | 4343.69 | 5564.64 | 6561.37 | 0.95 | 0.84 | 0.74 |
| Simuliidae | 325.72 | 1081.13 | 1977.75 | 0.07 | 0.16 | 0.22 |

**Table A2.** *Cont.*

| Discharge | WUA [m$^2$] | | | HHS | | |
|---|---|---|---|---|---|---|
| | 0.07 m$^3$ s$^{-1}$ | 0.26 m$^3$ s$^{-1}$ | 1.15 m$^3$ s$^{-1}$ | 0.07 m$^3$ s$^{-1}$ | 0.26 m$^3$ s$^{-1}$ | 1.15 m$^3$ s$^{-1}$ |
| **FFGs** | | | | | | |
| Grazers | 3708.03 | 5853.05 | 8109.54 | 0.81 | 0.88 | 0.91 |
| Shredders | 1998.90 | 3176.06 | 4522.08 | 0.44 | 0.48 | 0.51 |
| P. filter feeders | 975.95 | 2194.06 | 3554.71 | 0.21 | 0.33 | 0.40 |
| Detritus C.G. | 4076.37 | 6204.29 | 8452.97 | 0.89 | 0.94 | 0.95 |
| Predators | 3295.45 | 5154.41 | 7206.70 | 0.72 | 0.78 | 0.81 |
| **Comm. metrics** | | | | | | |
| Total abundance | 4143.03 | 6263.97 | 8523.34 | 0.91 | 0.95 | 0.96 |
| Total biomass | 3971.58 | 6068.91 | 8327.91 | 0.87 | 0.92 | 0.94 |
| Taxa richness | 4244.45 | 6309.89 | 8570.50 | 0.93 | 0.95 | 0.96 |
| Shannon index | 3218.10 | 4739.16 | 6454.89 | 0.70 | 0.72 | 0.73 |
| Simpson index | 3805.70 | 5629.09 | 7653.01 | 0.83 | 0.85 | 0.86 |
| **Resources** | | | | | | |
| CPOM | 509.03 | 853.80 | 1351.71 | 0.11 | 0.13 | 0.15 |
| FPOM | 4506.49 | 6420.34 | 8547.81 | 0.99 | 0.97 | 0.96 |

## Appendix C

**Table A3.** Macroinvertebrate taxa list from the Orino and the Lesgiüna.

| TAXA | Orino | Lesgiüna |
|---|---|---|
| Aeschinidae | x | |
| *Cordulegaster* spp. | x | |
| *Ephemera danica* | x | |
| *Habroleptoides confusa* | x | x |
| *Epeorus alpicola* | | x |
| *Epeorus sylvicola* | x | x |
| *Ecdyonurus* spp. | x | x |
| *Rhytrogena degrangei* | | x |
| *Rhytrogena semicolorata* | x | x |
| *Baetis alpinus* | x | x |
| *Baetis rhodani* | x | x |
| *Alanites muticus* | x | x |
| *Siphlonurus lacustris* | | x |
| *Leptoblephia* spp. early instar (?) | x | |
| *Capnia nigra* | | x |
| *Leuctra* spp. | x | x |
| *Isoperla* spp. | x | x |
| *Isoperla grammatica* | x | x |
| *Isoperla rivulorum* | x | x |
| *Perla grandis* | x | x |
| *Protonemoura* spp. | x | x |
| *Protonemoura nimborum* | x | x |
| *Amphinemoura sulcicollis* | x | x |
| *Nemoura mortoni* | | x |
| Limnephilidae | x | x |
| *Hydroptila* spp. | x | x |
| Silo/Lithax | x | x |
| *Hydropsyche* spp. | x | x |
| *Allogamus auricollis* | x | x |
| *Ryacophila* spp. | x | x |
| *Sericostoma* spp. | x | |
| *Polycentropus* spp. | x | |

**Table A3.** *Cont.*

| TAXA | Orino | Lesgiüna |
|---|---|---|
| *Drusus* spp. | | x |
| *Potamophylax* spp. | | x |
| *Philopotamus ludificatus* | | x |
| *Philopotamus montanus* | | x |
| *Philopotamus* spp. | | x |
| Odontoceridae | x | x |
| *Enoycila pusilla* | | x |
| Psychodidae | | x |
| Stratiomidae | x | |
| Thaumaleidae | | x |
| *Clinocera* spp. | x | x |
| *Prosimulium* spp. | x | x |
| *Simulium* spp. | x | x |
| *Dicranota* spp. | x | x |
| *Hexatoma* spp. | x | x |
| Tipulidae | x | |
| *Atherix marginata* | x | x |
| *Atherix ibis* | | x |
| Tanypodinae | x | x |
| Orthocladiinae | x | x |
| Tanytarsini | x | x |
| Chironomini | x | x |
| *Riolus* spp. | x | x |
| *Esolus* spp. | x | x |
| *Elmis* spp. | x | x |
| *Dryops* spp. | x | x |
| Coleoptera | x | |
| Planaria | x | x |
| Oligochaeta | x | x |
| Hydracaridae | x | x |
| Nematomorpha | x | x |
| Mollusc (bivalve) | x | |

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
