# Peer review of "Predicting Macroinvertebrate Responses to Water Abstraction in Alpine Streams"

_water, doi:10.3390/w13152121_

Round 1

Reviewer 1 Report

General Comments

This is a well-organized, well-written study of importance to stream ecology and river management. Furthermore, this is an interesting study that shows promise in ecohydrological research. I recommend the paper be published with minor revision, however I strongly urge the editors and authors to include a supplemental table or appendix listing all taxa identified for this study keeping with calls for publication of biodiversity data associated with ecological research.

Specific Comments

Materials and Methods

Please clarify that you have only two sample sites/reaches. Do you have two reaches with a set of sites within each reach? If so, then explain that to your readers (also, see comments below about the length of these reaches, etc.). Typically having only two reaches in a study may be grounds for rejecting a paper, but you collect a great deal of data within these reaches. Your manuscript would be improved if you provide more details on your study design and make explicit the methods used to sample discharge, FST/shear stress, and macroinvertebrates within each of these two reaches.

Please justify or include background methods on why the study reaches are 70-m long and on why you include 17 transects for the collection of physical/habitat features (as opposed to any other number of transects). Also, were the sites selected for discharge transects randomly selected within the constraint of having a “regular profile”? How many of these cross sections did you measure along the 70-m reaches? For example, did you have on cross section upstream, one in the middle of the reach, and one downstream? Were these cross sections related to the 17 transects listed above and/or were they related to the placement of FST hemispheres? Please describe the placement of FST hemispheres within the reaches and how these data were collected in greater detail, particularly since the placement of FST hemispheres relates to where you collected benthic macroinvertebrate samples.

The aim of this paper seems to relate variation in hydraulic stress to variation in macroinvertebrate taxa, FFGs, and community metrics, but I am curious as to why you did not include some other functional trait response variables such as flow preferences. Relating your flow data in predictive models to flow preference functional traits may produce some indicators useful for natural resource managers to use as per your discussion.

I suggest that you make Figure 3 larger, more readable. It was difficult to read online and in hard copy.

Results

Section 3.4. I am not sure whether you can call these concave curves unless this is a specific designation of the models themselves.

Author Response

We are grateful to both reviewers for the constructive comments, which improved the manuscript. We have addressed all points raised by the reviewers by changes in the manuscript as described in this response letter. We have added further changes to the text of the manuscript to increase clarity and conciseness. All changes to the manuscript are highlighted by tracked changes.

As the reviewer interpreted correctly, the aim of our study was not to cover all regional stream types but to study two representative alpine streams and their macroinvertebrate communities with very high spatial and taxonomic resolution. This was our choice for the trade-off of study effort (i.e. focusing on this level of resolution in the parameters measured) because it is not commonly done.

We fully agree on the importance of published biodiversity data and thank the reviewer for raising this point. We added a table in the appendix with the taxa list, and referred to it in the results section (Appendix C, Table C1).

We added more details to material and methods, to clarify the sampling scheme and to emphasize the rationale behind some decisions as highlighted by the reviewer. Following the reviewer’s comment, we included additional explanation on how the hemispheres were operated, and how discharge was measured.

We agree with the reviewer that the data collected could have been used to generate preference curves for flow-related traits and others. In our study, we decided to focus on novel aspects of physical habitat models to investigate if and how near-bed hydraulic conditions (measured as FST) shape the distribution of FFGs, and of their resources. Developing flow preference curves also for flow-related traits would be an exercise to confirm or refute published trait values in our streams, which was not our intention. Instead, we decided to specifically discuss such interpretations in the discussion.

We increased size and resolution of Figure 3, and added details in the caption to facilitate its interpretation.

We have changed wording of convex and concave curve to bell and bowl shaped.

We also edited the text in the introduction and the discussion as suggested by the reviewer. By going once again through the text, we also revised and streamlined some sentences.

Reviewer 2 Report

I think this is an interesting and well-written paper that reports on a well-designed study. My comments are minor.

Page 5, last paragraph: I'd start this paragraph by reminding the reader what the hydraulic model is for.

Page 6, last paragraph: I’d cut the first sentence, and insert “the three” prior to “different discharges.”

Page 7, last paragraph, first sentence: I’d replace “in-between” with “intermediate.” In the second sentence, insert “with increasing discharge” after “0 numbers.” Otherwise, the meaning of the sentence is very unclear.

Figure 3. I’ve never seen a figure quite like this, and it seems kind of inefficient. Each circle is a point estimate of the slope of a line? Why not show the full response curve rather than two arbitrary points? Also, please add some kind of panel labels (following journal style guidelines) so that individual panels can be referenced in the caption and in the text.

Page 10, last paragraph. “Maxima” should be “maximum” since it is singular here.

Author Response

We are grateful to both reviewers for the constructive comments, which improved the manuscript. We have addressed all points raised by the reviewers by changes in the manuscript as described in this response letter. We have added further changes to the text of the manuscript to increase clarity and conciseness. All changes to the manuscript are highlighted by tracked changes.

We edited the text in all instances as suggested by the reviewer to increase clarity. We also revised and streamlined other sections.

Concerning Figure 3, we agree with the reviewer that it contains a lot of information, which may challenge interpretations. We spent quite some time to find a way to effectively visualize a summary of the results of the physical habitat model. We tried with tables, but they were huge and did not effectively highlight changes in HSI and WUA with changing discharge (the tables can be found in Appendix B, Tables B1 and B2). We also tried to draw full response curves, but they were often crossing each other, and the result was quite chaotic and therefore difficult to interpret for the reader. We eventually decided to calculate deltas to effectively visualize changes in HSI and WUA. In this way, we had fewer lines crowding the figure, and almost no crossing. The two points displayed are not arbitrary; they represent the change in habitat suitability (HSI) and the area weighted by its habitat suitability (WUA) for a given variable: Δ1 between low and mid discharge, and the cumulative discharge increase Δ12 2 being the difference between mid and high discharge).

We opted for the cumulative delta instead of Δ2 alone to more clearly show the effect of a further increase in discharge. It can be interpreted as follows: If a point is above 0 (dashed line) in Δ1 it means that HSI or WUA increases with discharge going from low (baseflow) to mid flow. The slope of the line between the points for Δ1 and Δ12 indicates the magnitude of change due to Δ2. If the slope of the line is positive, it means that an increase in discharge from mid to high further increases HSI or WUA, while if the slope of the line between the two points is negative, it means that Δ2 has negative effects. If the slope is horizontal, it means that Δ2 does not have effects. If the line goes below the dashed line (zero change compared to low flow conditions) it means that the discharge increase from mid- to high flow causes a decline of HSI or WUA compared to the low-flow conditions. If a point is already below the dashed line in Δ1 it means that already increasing discharge from low to mid negatively affects HSI or WUA. In few cases WUA increases despite a decrease in HSI due to an increase in wetted area (described in the discussion). We increased size and resolution, added panel labels as suggested, and added these detailed explanations to the caption to facilitate the interpretation of the figure.